# Negative Regulation of the Differentiation of Flk2^−^ CD34^−^ LSK Hematopoietic Stem Cells by EKLF/KLF1

**DOI:** 10.3390/ijms21228448

**Published:** 2020-11-10

**Authors:** Chun-Hao Hung, Keh-Yang Wang, Yae-Huei Liou, Jing-Ping Wang, Anna Yu-Szu Huang, Tung-Liang Lee, Si-Tse Jiang, Nah-Shih Liao, Yu-Chiau Shyu, Che-Kun James Shen

**Affiliations:** 1Institute of Molecular Biology, Academia Sinica, Nankang, Taipei 115, Taiwan; libur777@gate.sinica.edu.tw (C.-H.H.); kywang@gate.sinica.edu.tw (K.-Y.W.); yliou@hsph.harvard.edu (Y.-H.L.); wjp976@gmail.com (J.-P.W.); annabellehuang7645@gmail.com (A.Y.-S.H.); andywork0711@gmail.com (T.-L.L.); mbfelix@imb.sinica.edu.tw (N.-S.L.); 2Department of Research and Development, National Laboratory Animal Center, National Applied Research Laboratories, Tainan 74147, Taiwan; stjiang@nlac.narl.org.tw; 3Department of Nursing, Chang Gung University of Science and Technology, Taoyuan City 333, Taiwan; 4Community Medicine Research Center, Chang Gung Memorial Hospital, Keelung Branch, Keelung 204, Taiwan; 5The PhD Program for Neural Regenerative Medicine, Taipei Medical University, Taipei 115, Taiwan

**Keywords:** Flk2-CD34-HSC, EKLF/KLF1, hematopoiesis, differentiation

## Abstract

Erythroid Krüppel-like factor (EKLF/KLF1) was identified initially as a critical erythroid-specific transcription factor and was later found to be also expressed in other types of hematopoietic cells, including megakaryocytes and several progenitors. In this study, we have examined the regulatory effects of EKLF on hematopoiesis by comparative analysis of E14.5 fetal livers from wild-type and *Eklf* gene knockout (KO) mouse embryos. Depletion of EKLF expression greatly changes the populations of different types of hematopoietic cells, including, unexpectedly, the long-term hematopoietic stem cells Flk2^−^ CD34^−^ Lin^−^ Sca1^+^ c-Kit^+^ (LSK)-HSC. In an interesting correlation, *Eklf* is expressed at a relatively high level in multipotent progenitor (MPP). Furthermore, EKLF appears to repress the expression of the colony-stimulating factor 2 receptor β subunit (CSF2RB). As a result, Flk2^−^ CD34^−^ LSK-HSC gains increased differentiation capability upon depletion of EKLF, as demonstrated by the methylcellulose colony formation assay and by serial transplantation experiments in vivo. Together, these data demonstrate the regulation of hematopoiesis in vertebrates by EKLF through its negative regulatory effects on the differentiation of the hematopoietic stem and progenitor cells, including Flk2^−^ CD34^−^ LSK-HSCs.

## 1. Introduction

Hematopoiesis is the process in which the hematopoietic/blood system generates multiple types of myeloid and lymphoid blood cells [1,2]. In mice, the primary blood system starts to produce blood cells from embryonic day (E) 7.5 in the blood island and then in yolk sac, which mainly contain erythrocyte, megakaryocyte, and macrophage [3]. The definitive blood system develops at E10.5 in arota-gonad-mesonephros (AGM), then in the fetal liver starting at E11, and finally in the bone marrow [4]. As in the adult bone marrow, the mouse fetal liver blood system also consists of multiple lineages, with the lymphoid lineage leading to T, B, and natural killer (NK) cells while the megakaryocyte, erythrocytes, granulocyte, and monocyte-macrophage belong to myeloid lineage. Lymphoid and myeloid lineage commitment occurs in multipotent hematopoietic progenitors, including the multipotent progenitor (MPP), the common myeloid progenitor (CMP), the myeloid/erythroid progenitor (MEP), the granulocyte/macrophage progenitor (GMP), and the common lymphoid progenitor (CLP), with MPP generated through self-renewal and differentiation of hematopoietic stem cells (HSC) [5,6,7,8,9].

HSCs primarily reside in the G0 phase under homeostatic conditions [10]. In murine, primitive HSCs could directly differentiate into mature red blood cells, but the definitive HSCs in late-stage fetal liver and bone marrow undergo multi-lineage differentiation pathways as described above [11]. Furthermore, a class of long-term self-renewing HSCs (LT-HSC) has been defined and isolated with use of different sets of surface markers, such as CD150^+^ CD48^−^ Lin^−^ Sca1^+^ Kit^+^/Flk2^−^ CD34^−^ Lin^−^ Sca1^+^ Kit^+^/CD34^−^ Lin^−^ Sca1^+^ Kit^+^/EPCR^+^ Lin^−^ Sca1^+^ Kit^+^/CD45^mid^ Lin^−^ Rhodamin^−/lo^ [2,12,13,14,15,16,17,18]. Being at the very top of the hematopoietic cellular system, HSCs play a major role in hematopoiesis and the regulation of their homeostasis determines the downstream fates of various hematopoietic/blood cells. Notably, a number of cytokines (such as IL-3, IL-7, SCF, TPO, GM-CSF, etc.) and transcription factors (such as Notch1, Tal-1, HoxB4, GATA1, GATA2, GATA3, etc.) are involved in the regulation of the homeostasis of HSCs and hematopoietic progenitors [6,19,20]. Furthermore, fetal liver niche factors, such as IGF-2, SCF, CXCL12, etc. [21], and endothelial-related antigens, including Tie2/angiopoietin receptor-2, vascular-endothelial cadherin, CD34, etc. [22], have been demonstrated to influence the homeostasis of HSCs. The morphologic and functional properties of purified HSCs have been extensively characterized. Furthermore, a number of studies have been reported on the regulation of the self-renewal, maintenance, and differentiation of HSCs at the molecular and cellular levels [19,23]. However, despite the above, much of the molecular and cellular basis of the regulation of hematopoiesis remains to be investigated.

EKLF/KLF1 is the first member of the Krüppel-like factor family consisting of an N-terminal activation domain, a C-terminal zinc finger domain, and multiple post-translational modification sites [24,25,26]. Genome-wide analysis of mouse fetal liver has identified a number of genes, the activation or repression of which are regulated through DNA-binding of EKLF to specific regulatory regions [27,28]. EKLF/KLF1 was identified initially as an erythroid-specific transcription factor but was later found to be also expressed in megakaryocyte and hematopoietic progenitors, including the MEP and GMP as well as the CMP [29]. Loss-of-function and gain-of-function studies have shown that EKLF not only regulates the process of erythropoiesis [30,31] but also the differentiation-fate decision from the MEP to erythrocyte or megakaryocyte [29]. Whether, and how extensively, EKLF participates in the regulation of hematopoiesis, other than the megakaryocyte–erythrocyte separation and monocyte-to-macrophage [30], has remained unknown.

In this communication, we report that EKLF is also expressed in LT-HSCs as defined by the marker set Flk2^−^ CD34^−^ Lin^−^ Sca1^+^ Kit^+^ (LSK) [12], which we term as Flk2-CD34-HSC throughout the studies below. We further show that EKLF balances hematopoiesis by negatively regulating the differentiation of hematopoietic stem- and progenitor cells (HSPCs), in part through repression of the expression of a common subunit of IL3/IL5/GM-CSF receptors in HSPCs. These findings establish EKLF as a hierarchical regulator of hematopoiesis in mammals, as suggested by the study of mouse fetal hematopoiesis shown below.

## 2. Results

### 2.1. Disturbance of Homeostasis of the Hematopoietic Cells upon Depletion of EKLF

To examine the regulatory effects of EKLF on the homeostasis of the hematopoietic system, other than the differentiation of erythroid vs. megakaryocyte lineages, we first generated a mouse model with *Eklf* gene-knockout (KO) using the conventional gene-targeting approach (Figure 1A,B). Similar to previous studies by others [24,32], the homozygous *Eklf*
^−/−^ mice were embryonic lethal at E14.5 and the mutant embryos were anemic, exhibiting albino-like phenotypes, in part due to the lack of globin gene expression. The EKLF expression level shown in immunoblotting (IB) and the genotyping strategical PCR are presented as well (Figure 1B). We then prepared E14.5 fetal livers from the E*klf*
^+/+^ (wild-type (WT)) and *Eklf*
^−/−^ mice (KO) and sorted the cells using a flow cytometer after staining with different combinations of antibodies.

The hematopoietic cells of WT and KO E14.5 fetal livers were analyzed in spite of the fact that *Eklf*-KO mice were embryonic lethal on E14.5. This was because of the larger size of the E14.5 fetal livers than E13.5 fetal livers and, consequently, more fetal liver cells for analysis. Of course, the E14.5 KO embryos were assured to be alive before dissection. As expected from previous studies [29], absence of EKLF led to a great loss of erythrocytes and concomitant increase in megakaryocytes in the E14.5 fetal livers of KO mice (Figure 1C and Appendix A). Concomitantly, we found that in KO E14.5 fetal livers, the absolute numbers of most types of hematopoietic cells, including Flk2^-^CD34-HSCs, MPPs, CMPs, GMPs, MEPs, macrophages, and dendritic cells, in the classical hierarchical model, were also changed in comparison to the WT E14.5 fetal livers. Among them, the Flk2-CD34-HSC number was significantly decreased while those of MPPs, CMPs, and GMPs were increased (Appendix A). In addition, dendritic cells were also increased but MEPs and macrophages were decreased. On the other hand, cells and monocytes remained relatively unchanged (Appendix A, Figure 1D,E).

The above data suggest that EKLF globally regulates the homeostasis of the hematopoietic system. In particular, the absence of the factor leading to the number decrease in Flk2-CD34-HSC and number increases in MPP/CMP/GMP suggests that EKLF negatively regulates the differentiation of Flk2-CD34-HSC into its downstream progenitors (also see below).

### 2.2. Expression Patterns of Eklf and Csf2rb in Hematopoietic Stem Cells and Progenitors

To elucidate the possible molecular basis of the regulatory effects of EKLF on hematopoiesis, we first analyzed and compared the levels of *Eklf* mRNA in Flk2-CD34-HSC and different hematopoietic progenitors. As shown by RT-qPCR analysis of mRNAs of WT E14.5 fetal liver, the *Eklf* mRNA level in MEP was comparable to that in the murine erythroleukemia (MEL) cells, which is an EKLF-high expressing erythroid cell line frequently used as a model system for studies of erythropoiesis in vitro [25,33], while those in CMP and GMP were fairly low (left histogram, Figure 2A). This pattern of *Eklf* expression was similar to that derived from analysis of MEP, CMP, and GMP isolated from adult mouse bone marrow [29]. Surprisingly, however, the levels of *Eklf* mRNA in MPPs as well as Flk2-CD34-HSCs of the E14.5 fetal liver were relatively high, approximately 50% of that of MEPs. In contrast, the Flk2-CD34-HSCs purified from mouse bone marrow exhibited very low levels of *Eklf* mRNA (left histogram, Figure 2A), as reported by others before [34]. As expected, *Eklf* mRNA was absent in the above types of cells of E14.5 fetal liver of KO mice, as exemplified for Flk2-CD34-HSCs and MPPs (right histogram, Figure 2A).

Homeostasis of the hematopoietic system depends, in part, on the balance of the self-renewal of Flk2-CD34-HSCs and proliferation of the different hematopoietic precursors with their differentiation capabilities, which are modulated by various cytokines and signal transduction pathways [35,36,37]. In view of the population changes of the hematopoietic cells in the E14.5 fetal livers of KO mice (Figure 1D,E), we carried out quantitative RT-qPCR analysis of the expression of three genes, *Csf2rb*, *Stat1*, and *Stat2*, known to be involved in the proliferation, self-renewal, and/or maintenance of hematopoietic stem cells and progenitors [38]. As shown in Figure 2B, the levels of *Stat1* mRNA and *Stat2* mRNA remained unchanged in CMPs and GMPs, but *Stat1* mRNA was increased in MPPs upon knockout of *Eklf*^-^. On the other hand, the level of *Csf2rb* mRNA, which encoded the common subunit CSF2RB of the IL-3/IL-5/GM-CSF receptors, was substantially increased in these three progenitors (Figure 2B). Since *Csf2rb* is highly expressed in mast cells [39] (database Bio-GPS: http://biogps.org/#goto=genereport&id=12983), we checked whether the upregulation of *Csf2rb* in the progenitors from KO E14.5 fetal livers was due to contamination by mast cells. There was little or no mast cell contamination in the hematopoietic stem/progenitor cells from WT or KO mouse fetal livers, as validated by RT-qPCR analysis of the mast-cell-specific *Fcer1a* mRNA expression.

We further analyzed the expression level of *Csf2rb* mRNA in Flk2-CD34-HSCs. As shown in Figure 2C, *Csf2rb* mRNA was increased in Flk2-CD34-HSCs as well upon depletion of EKLF. Fluorescence co-immunostaining showed that the level of CSF2RB protein was also elevated in Flk2-CD34-HSCs of the KO mouse E14.5 fetal liver (Figure 2D). Interestingly, EKLF-staining signals were present in the nucleus (arrows, Figure 2D) but the majority of EKLF was located in the cytosol of Flk2-CD34-HSCs (Figure 2D), a distribution pattern similar to that previously observed in erythroid progenitors [25].

### 2.3. Negative Regulation of the Multi-Lineage Differentiation of Flk2-CD34-HSC by EKLF

To further understand the basis of the decrease in Flk2-CD34-HSC numbers in the E14.5 fetal livers of *Eklf*
^−/−^ mice, we carried out the colony-forming cell (CFC) assay of sorted Flk2-CD34-HSCs as described by [40]. As expected, there was no colony formed when the sorting-purified Flk2-CD34-HSCs from either *Eklf ^+/+^* or *Eklf^-/^*^−^ E14.5 fetal livers were cultured in the cytokine-free methylcellulose medium on plates. However, when incubated with the cytokines/factors rmSCF, rhIL-6, and rmIL-3 in the absence of rhEPO, approximately 150 out of 10^3^ WT Flk2-CD34-HSCs would form colonies (left bar of the histogram, Figure 2E). Furthermore, the Flk2-CD34-HSCs from *Eklf ^−/−^* E14.5 fetal livers gained more robust differentiation capacity upon stimulation by the cytokines/factors, as reflected by the 2.5-fold increase in the colony number (right bar of histogram, Figure 2E).

Since the E14.5 *Eklf ^−/−^* fetal liver was anemic and under a hypoxia condition [41] that might affect the differentiation of Flk2-CD34-HSCs within [42], we used CRISPR/Cas9 editing to construct a mutant ESC clone, ESC(Del), in which the expression of EKLF was knocked out by targeted deletion of an internal region of the *Eklf* gene and consequent generation of a pre-mature termination codon (Figure 2F). Hematopoietic differentiation in vitro of the ESCs in a chemically defined medium and formation of the embryoid bodies (EB) were then carried out as described in the Materials and Methods. Cells that dissociated from the wild-type (WT) EB and the mutant EB were then subjected to the CFC assay. As shown in Figure 2G, the definitive hematopoietic cells derived from the mutant ESC(Del) after differentiation in culture for 14 days [43,44] formed a significantly higher number of colonies than those derived from WT ESC. The data of Figure 2E,G together indicate that under normal conditions, EKLF maintains the homeostasis of Flk2-CD34-HSCs and HSPCs, in part, through prevention of their over-differentiation into downstream hematopoietic cells.

### 2.4. Enhancement of Myeloid Differentiation, and Self-Renewal Capability of Flk2-CD34-HSC (KO) and HSPC (KO) in Transplanted Mice

To investigate the self-renewal and differentiation properties of Flk2-CD34-HSC in vivo, we carried out a serial transplantation assay of E14.5 fetal liver cells from the WT and KO mice using the CD45.2 congenic marker for tracking the donor origin (Figure 3A). For the primary transplantation, mouse fetal liver cells (5 × 10^6^) were injected into the tail vein of CD45.1 WT mice, as described, e.g., by Tiago et al (2009). The injection dose of 5 × 10^6^ cells/mouse was based on the survival rates of mice injected with different doses.

As shown in Figure 3B, around 90% of the peripheral blood cells in the CD45.1-recipient mice were CD45.2, whether the transplanted CD45.2 cells were from WT or KO fetal liver. Notably, KO fetal liver-derived leukocytes exhibited a significantly higher population (80%) than that (~50%) of WT fetal liver cell-derived leukocytes in the peripheral blood of tertiary recipient mice (Figure 3B). It should be mentioned here that there was a great decrease in the CD45.2 chimerism of WT cells in tertiary recipients (Figure 3B). We suspect that it might be due to HSPC exhaustion [45]. Furthermore, while the peripheral blood populations of donor-derived T cells (CD3^+^), B cells (B220^+^), monocytes (CD11b^+^ Gr1^−^), and granulocytes (CD11b^+^ Gr-1^+^) exhibited no significant differences between the primary or secondary recipient mice of WT and KO fetal liver cells, KO fetal liver cell-derived monocytes (CD11b^+^ Gr1^−^) and granulocytes (CD11b^+^ Gr-1^+^) greatly increased and T cells (CD3^+^) decreased in the tertiary recipients (Figure 3C).

In an interesting parallel, nearly 100% of the CD45.1/CD45.2 mononuclear cells, including Flk2-CD34-HSCs and LSK cells in the bone marrow (BMMNC) of CD45.1-recipient mice, were marked with CD45.2 after the primary transplantation (top of histogram, Figure 4A). After the secondary transplantation, however, the percentages of CD45.2 LSK(KO) cells and CD45.2 Flk2-CD34-HSCs(KO), but not CD45.2 LSK(WT) cells and CD45.2 Flk2-CD34-HSC(WT), in the bone marrow of CD45.1-recipient mice became higher, while those of the CD45.2-positive BMMNCs remained similar (99.3% for WT and 97.9% for KO) as in the primary recipient mice (compare the three histograms in Figure 4A). Notably, similar to some other studies using the serial transplantation assay [46,47], there was a magnification of the pool size of non-CD45 BMMNCs in the bone marrow of primary, secondary, and tertiary recipients of KO fetal liver cells (Figure 4A), which might be early-stage cells of the Flk2-CD34-HSC hierarchy or stem cells from the embryonic stage [48,49]. Significantly, BMMNCs and LSK(KO) cells as well as Flk2-CD34-HSCs (KO) in the bone marrow of CD45.1-recipient mouse all displayed a higher percentage of donor marker (CD45.2) after the tertiary transplantation (Figure 4A).

## 3. Discussion

Appropriate hematopoiesis is pivotal for normal production and functioning of the blood system. Taking advantage of our genetically engineered mouse model (*Eklf*
^−/−^) lacking the expression of EKLF (Figure 1), we demonstrate, in this study, that this DNA-binding transcription factor plays an essential regulatory role in the differentiation of Flk2-CD34-HSCs and hematopoietic progenitors in the myeloid lineage of mice. Evolutionarily conserved in vertebrates, EKLF has been known to positively regulate erythropoiesis and participates in the differentiation-fate decision of MEPs into erythrocytes vs. megakaryocytes [24,29]. For the first time, we show that depletion of EKLF leads to population changes in hematopoietic stem and progenitor cells—in particular, decreases in Flk2-CD34-HSC and MEP populations and increases in those of MPP, CMP and GMP in E14.5 mouse fetal liver (Figure 1). This suggests that EKLF plays an essential regulatory role in hematopoiesis by balancing the mono-myeloid lineage.

An interesting correlation with EKLF regulating the homeostasis of the myeloid lineage is its expression in not only MEPs, CMPs, GMPs, and MPPs but also in Flk2-CD34-HSCs (Figure 2A). A high level of *Eklf* mRNA in MEPs but not in other hematopoietic progenitors has been reported previously by [29]. However, we have found that the level of *Eklf* mRNA is also relatively high in MPP, approximately 50% of that in MEP. Furthermore, *Eklf* is expressed in the mouse Flk2-CD34-HSC (Figure 2A). These differences might be due to the different tissues analyzed and different sets of surface markers used for sorting of MPP and Flk2-CD34-HSC. In particular, Fontelo et al. have used Flk2^-^ as one of the markers for sorting MPP from adult mouse bone marrow cells [29], while we have used Flk2^+^ following the studies by Yang et al. and Adolfsson et al. to sort MPP from E14.5 fetal liver [18,50]. Furthermore, the combination of Lin^-^ Scal^+^ Kit^+^ (LSK) Thyl.1^lo^ Flk2^−^ was used in the previous study to sort bone marrow Flk2-CD34-HSCs. We, instead, have included CD34^-^ as one of the selection markers for sorting Flk2-CD34-HSCs from E14.5 fetal livers. Notably, our data are in parallel to the findings of *Eklf* expression in human iPSC-derived hematopoietic precursor cells [8] as well as in HSCs from both human and mouse (database Bio-GPS: http://biogps.org/#goto=genereport&id=16596). In addition, the up-regulation of EKLF in HSCs had been observed under non-physiological conditions, such as TET knockout/DNMT3A knockout [51] and perturbations of the signaling pathway of IKK2/NF-κB [52].

Mechanistically, depletion of EKLF up-regulates expression of the receptor subunit *Csf2rb* in Flk2-CD34-HSCs, MPPs, CMPs, and GMPs at the mRNA level (Figure 2B,C) and likely at the protein level as well, as shown for Flk2-CD34-HSCs (Figure 2D). *Csf2rb* encodes the common beta chain of the high-affinity receptors of IL-3, IL-5, and GM-CSF [53]. Of these cytokines, GM-CSF, functions as a hematologic cell growth factor, stimulating blood stem cells to generate granulocyte and monocyte lineages [54]. It also protects immunity, mainly by stimulating the maturation of dendritic cells [55]. Cancer immunotherapy takes advantage of tumor cell killing by mature dendritic cells and macrophages, which are stimulated or recruited by GM-CSF [56]. On the other hand, IL-3 and IL-5 are predominantly produced by T cells and they are required for not only differentiation but also proliferation and survival of the myeloid precursor cell CMPs, GMPs, and MEPs [57,58]. Finally, *Csf2rb* has been reported, previously, to be expressed in HSC and it is required for differentiation of HSCs [38]. Indeed, in part through release of the transcriptional repression of *Csf2rb* (Figure 2C,D), either Flk2-CD34-HSCs(KO) isolated from E14.5 *Eklf ^−/−^* fetal liver (Figure 2E) or HSPCs derived from CRISPR/Cas9-edited ESCs (Figure 2G) have enhanced differentiation capability upon the depletion of EKLF.

The effects of knockout of EKLF on the homeostasis of Flk2-CD34-HSC in vivo have been assayed by serial transplantations (Figure 3). As shown, in parallel to the analysis of the E14.5 fetal liver cells (Figure 1 and Figure 2), depletion of EKLF also causes population changes of the different types of peripheral blood cells after tertiary transplantation, with a favoritism towards the myeloid lineage (Figure 3C). The absence of EKLF also appears to allow the maintenance of a higher pool size of Flk2-CD34-HSCs, LSK cells, and BMMNCs in the bone marrow of tertiary recipient mice of KO E14.5 fetal liver cells than those transplanted with the WT E14.5 fetal liver cells (Figure 4A). However, since anemic conditions would promote the self-renewal of stem cells, including HSCs, while inhibiting their differentiation [59], the apparently higher self-renewal capability of Flk2-CD34-HSCs(KO) in the serially transplanted mice could be the result of their anemic nature due to impaired erythropoiesis of the KO blood. Furthermore, the HSC (KO) phenotypes might be in part due to some non-cell autonomous mechanisms [31].

In sum, our results have established that under normal conditions, EKLF participates in the balancing of the homeostasis of the hematopoietic/blood system by negatively regulating the differentiation capability of Flk2-CD34-HSCs. While a complete picture of the molecular basis of this regulation awaits to be examined, our data strongly suggested that EKLF suppresses IL3/IL5/GM-CSF-stimulated proliferation and differentiation of MPPs, CMPs, and GMPs as well as the differentiation of Flk2-CD34-HSCs by repressing the transcription of *Csf2rb*, directly or indirectly, thus maintaining the appropriate pool size of self-renewable Flk2-CD34-HSCs in a positive way (see model in Figure 4B). Whether EKLF also regulates the self-renewal capability of Flk2-CD34-HSCs awaits further investigation.

## 4. Materials and Methods

### 4.1. Generation of Eklf-KO Mice

C57BL16, or B6, mice (Jackson Laboratory, Bar Harbor, ME, USA) were used throughout the study. The generation of B6 mouse lines with heterozygous and homozygous knockout of the *Eklf* gene was carried out in the Transgenic Core Facility (TCF) of IMB, Academia Sinica, following the standard protocols. BAC construct containing genetically engineered *Eklf* locus (for more details, see the legend of Figure 1A) and E2A-Cre mice were used for the generation of the *Eklf*-KO mice. This study was conducted with the approval of the Institutional Animal Care and Utilization Committee (IACUC) (ID: 12-04-360; 17-02-1052), Academia Sinica (Taipei, Taiwan) and the protocol approval date were from 05 May 2012 to 11 January 2018 and 24 April 2017 to 06 October 2020. For PCR-based genotyping, 50 bp deletion (gray block) was introduced into intron 1 after the 5′ end of the LoxP site. The locations of the PCR primers used for genotyping are shown as small black arrows: 5’-deletion: 5′-GCG GCG CGA TAA CTT CGT AT-3’, 5’-PGK: 5′-TTG AAT TCT GCT TCC TGT TGG A-3’, EKLF-F: 5′-AGG CAG AAG AGA GAG AGG AGG C-3’, 3’-deletion: 5′-CCT ATT TCT CCA ACA GGA AGC A-3’, PGK-R: 5′-CTG GCC CTC AAA CAA CCC TG-3′, 3’-PGK: 5’-GTT ATG CGG CCC TAG TGA TTT A-3’. Nifx and Fbwx9 are two distal gene loci flanking the *Eklf* locus.

### 4.2. Flow Cytometric Analysis and Cell Sorting

WT and KO E14.5 fetal liver cells were filtered through a 40-mm nylon cell strainer (BD Biosciences, San Diego, CA, USA) to obtain single-cell suspension before flow sorting. Different types of mouse peripheral blood cells, including T cells, B cells, granulocytes, and monocytes (see Appendix A), were subjected to flow sorting after RBC lysis buffer treatment of the peripheral blood of 2-month-old mice. Bone marrow (BM) Flk2-CD34-HSC of 2-month-old mice was isolated by flow sorting the bone marrow mono-nuclear cells (BMMNCs) obtained by Ficoll density gradient centrifugation. As listed in Appendix A, different types of hematopoietic cells were identified with use of different combinations of the following antibodies against cell surface markers: anti-Lin, anti-Sca-1, anti-c-Kit (CD117), anti-CD34, anti-Thy1.1, anti-Flk2, anti-CD16/32, anti-CD11b, anti-CD11c, anti-Ter119, anti-CD42d, anti-CD41, anti-Gr-1, anti-F4/80, anti-33D1, and anti-FceRI (BD Biosciences and Bioscience). After immunostaining with the antibodies, the cells were either analyzed with LSRII (BD Biosciences) and FlowJo software (V10, Tree Star, Franklin Lakes, NJ, USA) or sorted with FACSAriaII SORP (BD Biosciences).

It should be noted here that since EKLF is required for the expression of the erythroid-specific marker Ter119 among others [60,61], as well as for progression of the erythroid lineage passing the Pro-E stage [26], Ter119 alone could not be used in flow sorting to separate the erythroid progenitors from other types of hematopoietic cells in the KO E14.5 fetal liver. Thus, Sca1^+^ and CD34^+^ were used for sorting of Flk2-CD34-HSC (KO)/MPP (KO) and CMP (KO)/GMP (KO), respectively, without contamination by the early erythroid progenitors [2,13]. On the other hand, however, the possibility of sorted MEP (KO) being contaminated by the early erythroid progenitors, e.g., CFU-E, BFU-E, and Pro-E, in KO and WT E14.5 fetal livers could not be excluded. For this reason, MEP (KO) cells were not subjected to further gene expression analysis, such as that of *Csf2rb* (Figure 2B).

### 4.3. RNA Analysis

Total RNAs from murine E14.5 fetal livers and MEL cells were extracted with the TRIzol reagent (Invitrogen, Carlsbad, CA, USA). Micro-scale RNAs of the individually purified cells, including Flk2-CD34-HSCs, MPPs, CMPs, GMPs, and MEPs from fetal liver and Flk2-CD34-HSCs from bone marrow (BM-Flk2-CD34-HSC), were isolated using the RNAqueous-Micro Kit (Ambion, Austin, TX, USA). cDNAs were then synthesized using SuperScript II Reverse Transcriptase (RT) (Invitrogen) for RT-qPCR analysis. Quantitative real-time PCR (qPCR) analysis of the cDNAs was carried out with the LightCycler^®^ 480 SYBR Green I Master (Roche Life Science, Penzberg, Germany) and the products were detected by the Roche LightCycler LC480 Real-Time PCR instrument. The sequences of the primers used for the qPCR analysis were either home-designed, as shown in Appendix A, or downloaded from the online database PrimerBank: http://pga.mgh.harvard.edu/primerbank.

### 4.4. Immunofluorescence Staining Analysis

Flk2-CD34-HSCs purified by flow sorting, as described above, were suspended and fixed by 1% paraformaldehyde on 4-well culture slide (Millipore Millicell EZ SLIDE), permeabilized with 0.1% (*v*/*v*) Triton X-100, and stained with mouse anti-mouse CSF2RB (GeneTex, Irvine, CA, USA) or home-made rabbit anti-mouse EKLF [62]. Anti-mouse and anti-rabbit secondary antibodies were conjugated with Alexa Fluor 488 and 543, respectively. Additionally, 49-6-diamidino-2-phenylindole (DAPI) (Invitrogen) was used for staining of the nucleus. Fluorescence excitation and image expression were achieved using LSM710 and LSM510. Image data were analyzed by Image J software.

### 4.5. Generation of Mouse ES Cells (ESC) with Knockout of Eklf Expression by CRISPR/Cas 9

Internal deletion of the *Eklf* gene was conducted with help from the RNAi Core of Academia Sinica. A guide RNA (gRNA) sequence in the 5′-(N)_20_NGG-3′ framework targeting exon 2 of the *Eklf* gene without redundancy in the genome was identified and used. The *Eklf* gRNA expression plasmid was constructed by ligation of the DNA oligos encoding the *Eklf* gRNA into the BsmBI double-cutting sites of the pT7/tRNA(Gln)-sgRNA-pPuro vector. ES cells were then transfected with eSpCas9(1.1) (a gift from Feng Zhang, Addgene plasmid #71814) [63] and the *Eklf* gRNA expression plasmid. After selection by puromycin (2 μg/mL for 2 days) and recovery for 1 week, the individual ESC clones were picked and screened by PCR-sequencing for the ones carrying homozygous frame-shift mutations in the *Eklf* gene. The targeted deletion of *Eklf* exon 2 in the selected clone, ESC(Del), was confirmed by DNA sequencing (Figure 2F). Sequences of the DNA oligos for *Eklf* gRNA expression plasmid construction and PCR primers for sequence validation are listed in Appendix A.

### 4.6. Colony-Forming Cell (CFC) Assay

Colony formation on methylcellulose was carried out as described by Miller and Lai [40]. Fluorescence-activated cell sorter (FACS)-purified Flk2-CD34-HSCs from mouse fetal liver were cultured overnight in a stem cell culture medium (Serum-Free Expansion Media, STEMCELL, Vancouver, BC, Canada). The cells were then transferred to GF M2534 (STEMCELL) with the addition of rmSCF/rhIL-6/rmIL-3 and plated in petri dishes. Unlike Pilon et al. [61], we did not use the Methocult GF M3434 medium, which contained EPO. The number of colonies formed was counted 14 days after the plating.

For CFC assay of hematopoietic cells derived from ESCs, mouse ESCs were maintained in a methylcellulose-based medium (M3120, STEMCELL) supplemented with MTG (M6145, Sigma, St. Louis, MO, USA), SCF (78064, STEMCELL), FBS, and L-Glutamine (07100, STEMCELL). After 7 days, the cultures were further supplemented with FBS, MTG, SCF, IL-3 (78042, STEMCELL), and IL-6 (78050, STEMCELL) and cultured for another 1 week. The embryoid bodies (EB) formed were then trypsinized and the dissociated cells were re-suspended (10,000 cells/mL) in media (M3120, STEMCELLS) supplemented with L-Glutamine, MTG, BIT9500 serum substitute, SCF, IL-3, and IL-6 and plated in petri dishes. The colonies were identified and counted 14 days later.

### 4.7. Serial Transplantation Assay

For serial fetal liver transplantation experiments [64,65], we injected 5 × 10^6^, 3 × 10^6^, or 2.5 × 10^6^ E14.5 fetal liver cells from CD45.2^+^ donor embryos into the tail veins of lethally irradiated (10 Gy), CD45.1^+^ C57BL/6 mouse recipients. Ten weeks later, 2 × 10^5^ BMMNCs from the primary recipients were transferred into lethally irradiated CD45.1^+^ recipient mice. After another 10 weeks, 1 × 10^6^ BMMNCs from the secondary recipient mice were transferred into lethally irradiated CD45.1^+^ tertiary recipients. The reconstituted hematopoietic lineages in the peripheral blood and bone marrows of the recipient mice were analyzed at 8 weeks and 10 weeks, respectively, post-transplantation.

Note that although the recipient mice received lethal doses of irradiation and the reconstituted hematopoietic/blood systems from the transplanted KO E14.5 fetal liver cells would not generate mature erythroid cells, most of them survived until around 12 weeks post-transplantation.

### 4.8. Statistics

Significant differences were determined using a two-tailed Student’s *t*-test (Microsoft Excel). *p* values ≤ 0.05 were considered significant.

## Figures and Tables

**Figure 1 ijms-21-08448-f001:**
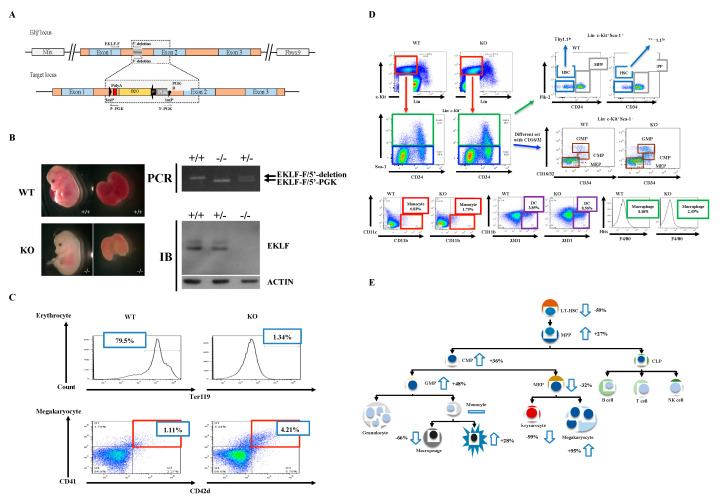
Generation of mice with gene knockout (KO) of *Eklf* and population changes in the myeloid lineage cells in *Eklf*
^−/−^ E14.5 fetal liver. (**A**) Targeting strategy. The schematic diagram shows the genetic context of *Eklf* locus and the map of the targeting BAC construct harboring an inverted loxP-PGK-gb2-neo-loxP cassette in the intron 1 region of the *Eklf* gene. Note: neo, neomycin resistance gene; PGK, phosphoglycerate kinase I promoter; black arrow, the prokaryotic promoter gb2; black arrow heads, loxP sites. (**B**) Left panels, anemic phenotype of the homozygous *Eklf*
^−/−^ (KO) E14.5 embryo in comparison to the wild-type (WT) E14.5 embryo. Right upper two panels, genotyping of E14.5 embryos. Only a representative gel pattern of PCR using the primers EKLF-F/5′-deletion for wild-type and EKLF-F/5′-PGK for the mutant is shown here. +/+, wild-type; +/−, heterozygous *Eklf*
^+/−^; −/−, homozygous *Eklf*
^−/−^. Right lower two panels, immunoblotting (IB) analysis showing the depletion of EKLF protein expression by *Eklf* gene knockout. β-actin was used as an internal control. (**C**) Comparative fluorescence-activated cell sorter (FACS) analysis of erythrocytes and megakaryocytes in E14.5 fetal liver cells of the WT and *Eklf*
^−/−^ mice. Note the decrease in Ter119^+^ cells of erythroid lineage and increase in megakaryocytes in the *Eklf*
^−/−^ E14.5 fetal liver. *n* = 3. (**D**) FACS analysis using different combinations of antibodies to identify Flk2-CD34-hematopoietic stem cells (HSCs), the multipotent progenitor (MPP), common myeloid progenitor (CMP), granulocyte/macrophage progenitor (GMP), and myeloid/erythroid progenitor (MEP) in WT and *Eklf*
^−/−^ (KO) mouse E14.5 fetal liver. The differentiated cells were identified as the following: monocytes, dendritic cells, macrophages. The flow data for granulocytes are not shown here. *n* ≥ 6. The absolute number of cells per WT E14.5 fetal liver was approximately twice that of the KO E14.5 fetal liver (see Appendix A). (**E**) Cartoon chart showing the differentiation diagram of hematopoiesis and changes in the numbers of different types of hematopoietic cells in the KO fetal liver in comparison to the WT fetal liver. The change is defined as the increase (upward arrow) or decrease (downward arrow) in the average number of a specific type of blood cells in KO fetal liver relative to the WT fetal liver. For the absolute numbers per fetal liver of the different types of the cells, see Appendix A. *n* ≥ 6.

**Figure 2 ijms-21-08448-f002:**
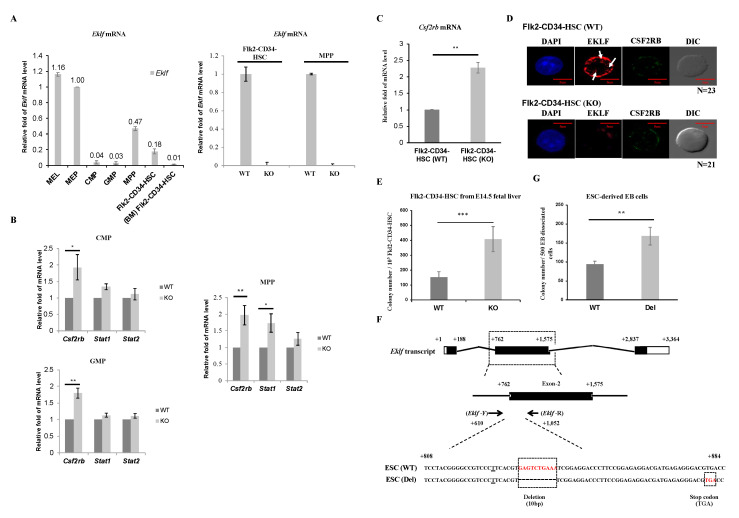
Regulation of Flk2-CD34-HSC differentiation by EKLF (**A**) RT-qPCR analysis of RNAs isolated from different types of mouse hematopoietic stem cells and progenitors, including LT-HSCs, MPPs, CMPs, GMPs, and MEPs purified from E14.5 mouse fetal liver and Flk2-CD34-HSCs from bone marrow (BM Flk2-CD34-HSC) by FACS. The relative levels of *Eklf* mRNA in all of the progenitor cells are compared to that of mouse erythroleukemia (MEL) cells in the left panel, with the level in MEP set as 1. The comparative RT-qPCR analyses of *Eklf* mRNA in Flk2-CD34-HSCs and MPPs of E14.5 fetal livers from WT and KO (*Eklf*
^−/−^) embryos are shown in the right panel. (**B**) RT-qPCR analysis of the mRNA levels of *Csf2rb*, *Stat1*, and *Stat2* in purified CMPs, GMPs, and MPPs. Five biological replicates were analyzed for each type of cells. Each bar represents mean ± standard deviation. * *p* < 0.05, ** *p* < 0.01. (**C**) Relative levels of *Csf2rb* mRNA in Flk2-CD34-HSC purified from WT and KO E14.5 mouse fetal livers were analyzed by RT-qPCR. *n* ≥ 5. ** *p* < 0.01. (**D**) Representative immuno-fluorescence staining patterns of the expression patterns of EKLF and CSF2RB in Flk2-CD34-HSCs (WT) and Flk2-CD34-HSCs (KO). DAPI (49-6-diamidino-2-phenylindole) is the nucleus marker. Note the lack of EKLF signal in Flk2-CD34-HSCs (KO). Three biological replicates were analyzed. The diameters of LT-HSCs range from 5–10 μm. EKLF signal present in the nucleus (white arrow). (**E**) Colony-forming cell (CFC) assay was performed on Flk2-CD34-HSCs (WT) and Flk2-CD34-HSCs (KO) purified (>90%) from E14.5 fetal liver cells by flow cytometry. *n* ≥ 4. *** *p* < 0.001. (**F**) Top, map of *Eklf* transcript with 3 exons. Middle, *Eklf* exon 2 as the target of gDNA. Bottom, comparison of the genomic sequence of the CRISPR/Cas9-edited Eklf exon 2 region in ESC (Del) to the wild-type sequence in ESC (WT). The *Eklf*-F and *Eklf*-R PCR primers used for validating the sequences are indicated. Note the presence of 10 bp deletion in ESC(Del) that causes a frame shift and consequent pre-mature termination TGA codon. (**G**) CFC assay of embryoid body (EB) cells derived from wild-type ESC(WT) and from CRISPR/Cas9-edited ESC, i.e., ESC(Del), respectively. *n* = 3. ** *p* < 0.01.

**Figure 3 ijms-21-08448-f003:**
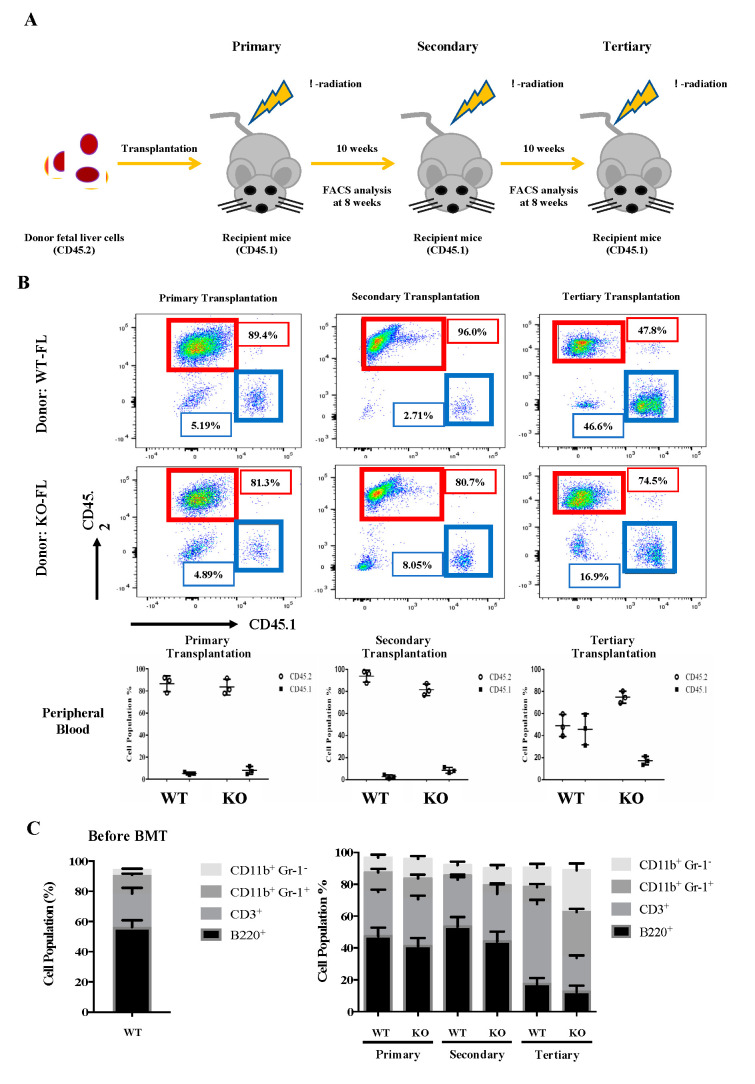
Serial transplantation of fetal liver cells from WT and *Eklf*
^−/−^ E14.5 embryos. (**A**) Strategy of the experiments. The fetal liver cells (CD45.2) were transplanted into lethally irradiated recipient mice. The percentages of donor/recipient chimerism of the peripheral blood were analyzed by flow cytometry at 8 weeks after each transplantation. (**B**) Flow cytometric analysis of the donor/recipient chimerism of the peripheral blood of the recipient mice. Representative FACS plots are shown in the upper six panels. Statistical analysis is shown in the lower three histograms. The different cell populations were normalized to the total number of CD45.1 and CD45.2 cells. The data represent mean ± S.D (primary transplantation, *n* = 3; secondary transplantation, *n* = 4; tertiary transplantation, *n* = 8) (**C**) Percentages of donor-derived lineage repopulations of T cells (CD3^+^), B cells (B220^+^), monocytes (CD11B^+^ Gr1^−^) and granulocytes (B11b^+^ Gr-1^+^) in the peripheral blood of recipient mice. The data represent mean ± S.D (primary transplantation, *n* = 3; secondary transplantation, *n* = 4; tertiary transplantation, *n* = 8).

**Figure 4 ijms-21-08448-f004:**
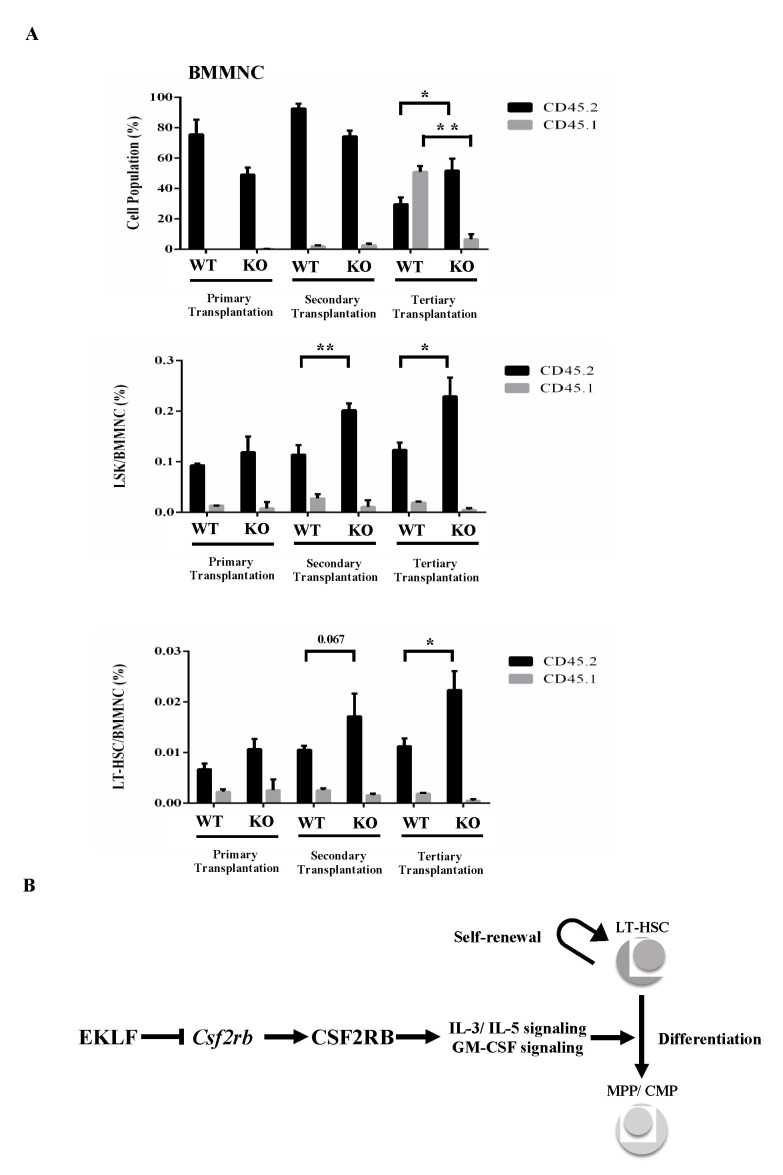
(**A**) FACS analysis of BMMNCs, Lin^−^ Sca1^+^ c-Kit^+^ (LSK) cells, and Flk2-CD34-HSCs in the bone marrow of recipient mice after serial transplantation. The bone marrow cells were stained with the appropriate antibodies and analyzed by FACS. Statistical analysis of the FACS data. The percentages of BMMNCs, ratios of LSK cells/BMMNCs, and ratios of Flk2-CD34-HSCs/BMMNCs in the bone marrows of primary, secondary, and tertiary recipient mice, as deduced from the FACS data, are shown in the histograms (primary transplantation, *n* = 3; secondary transplantation, *n* = 4; tertiary transplantation, *n* = 8). * *p* < 0.05, ** *p* < 0.01 (**B**) A simple model showing the negative regulatory roles of EKLF in the homeostasis of Flk2-CD34-HSCs. The factor restricts the self-renewal capability of Flk2-CD34-HSCs. It also acts as a repressor to prevent superfluous *Csf2rb* expression and consequent over-differentiation of Flk2-CD34-HSCs. For more details, see text.

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
