# Peer review of "Negative Regulation of the Differentiation of Flk2 CD34 LSK Hematopoietic Stem Cells by EKLF/KLF1"

_ijms, 2020, doi:10.3390/ijms21228448_

Round 1

Reviewer 1 Report

In this manuscript, Hung and colleagues take advantage of a self-made knock-out mouse for EKLF/KLF1 gene to claim its role inhibiting hematopoietic stem cell (HSC) differentiation in fetal liver. EKLF/KLF1 is well known as a master regulator of erythroid differentiation and multiple EKLF/KLF1 null model have been already generated. The potential novelty of this study resides on the fact that authors focus on primitive HSC instead of committed progenitors (megakaryocytic-erythroid progenitors). The manuscript is overall clear and well-structured; however, there are some points that require further discussion considering the controversy with previous publications: expression of EKLF/KLF1 in LT-HSC. More specific comments are found below.

Major comments:

1) The expression of EKLF/KLF1 by HSC has been reported up-regulated in non-physiological conditions (e.g in DNMT3A and TET2 KO HSCs, see Zhang, X. et al. Nat. Gen. 2016; or IKK2 activation, see Nakagawa, MM. et al. Cell Report. 2018). However, as discussed in this manuscript, healthy HSCs were shown to do not express EKLF/KLF1 (Frontelo, P. et al. Blood. 2007). This issue was justified based on different cell sources and markers used to identify LT-HSC. However, authors did not use the most updated and standard set of markers to identify primitive HSCs, which includes SLAM markers (see Oguro, H. et al. Cell Stem Cell. 2013). Therefore, it is difficult to assess if sorted cells are pure LT-HSCs or they are mixed with MPP, which are known to express EKLF/KLF1. Authors might consider to refer to CD34-Flk2- HSC instead of LT-HSC in order to bypass this controversy.

2) Considering the previously mentioned role of EKLF/KLF1 in non-physiological HSC, the authors should consider to highlight this aspect in the discussion and cite the corresponding studies.

3) In the introduction, sometimes it is not clear if authors are describing general concepts of adult bone marrow hematopoiesis or fetal liver hematopoiesis. Since the latter is the subject of this study, authors might provide some references of studies targeting other molecular/cellular mechanisms specifically in fetal liver hematopoiesis. Similarly, they should clearly state that their claim related to the role of "EKLF as a hierarchical regulator of hematopoiesis in mammals" applies so far to fetal hematopoiesis.

4) There are some evidences suggesting that not all EKLF effects are cell autonomous (Porcu, S. et al. Mol Cell Biol. 2011). This might explain why lethally-irradiated recipient mice transplanted with EKLF/KLF1 KO cells manage to survive till around 12 weeks post-transplantation. Since authors use a full KO model in their experiments, they should recognise that HSC phenotype might be partially due to non-cell autonomous mechanisms.

5) In general figure legends are extremely dense and show lot of information that properly belongs to the Material and Method section, or even to the results section. Please leave the essential information to understand the figures. Also check spaces between words after dots.

6) Authors should indicate why mouse erythroleukemia (MEL) cells are used as control for EKLF/KLF1 expression analysis.

7) Images provided in Fig. 2C are very small and it is actually difficult to distinguish any difference between WT and KO in terms of CSF2RB expression. Possibly authors could save space omitting merge images since DAPI masks the signal from other channels. In this regard, what do the authors mean when they state that "EKLF staining signals were present in the nucleus" (L.215).

8) Authors claim that EKLF/KLF1 KO exhibits enhanced long-term engraftment potential, but hematopoietic reconstitution just 2 months after transplantation is not sufficient for this claim. A minimum of 4 months after transplantation are required for it. Since these mice die after 12 weeks, authors should consider to focus on the self-renewal potential capability assessed by serial transplantations and ignore the effect on the long-term engraftment potential.

9) Since the amount of CD45.1- CD45.2- cells significantly changes along serial transplantations (Fig. 3B). Authors should consider to normalize their percentages to the total number of CD45.1 + CD45.2 cells they detect (taking them as 100%). Do they have any explanation to the dramatic decrease in the CD45.2 chimerism of WT cells in tertiary recipients? This is important considering that it increases slightly from primary to secondary recipients.

10) Authors fairly recognise that the self-renewal phenotype might be an indirect effect caused by the anemia. However, in the following paragraph they describe it as a consequence of the reduced differentiation capacity of KO HSC, and finally they claim that the role of EKLF on HSC self-renewal still needs to be elucidated (L. 349-354). Please be consistent with the claims.

Minor comments:

1) Please use the same font size and style for the current address information presented in the affiliations section.

2) The right nomenclature is Flk2, not F1k2.

3) L54-56: Provide reference for this claim.

4) There is no mention in the text to the PCR and IB gels shown in Fig. 1B.

5) L96-97: Please clarify what the authors mean with "the hematopoiesis in which was definitive in nature (Lee et al, 2016)".

6) Please avoid showing unnecessary mathematical calculations in the main text (e.g page 8). They just add confusion when single numbers are enough.

Author Response

Dear Dr. Mo,

We have revised our manuscript following the two reviewers' comments and suggestions, which were insightful and very helpful.

The point-to-point replies/ revisions are detailed below.

Major

1) The expression of EKLF/KLF1 by HSC has been reported up-regulated in non-physiological conditions (e.g in DNMT3A and TET2 KO HSCs, see Zhang, X. et al. Nat. Gen. 2016; or IKK2 activation, see Nakagawa, MM. et al. Cell Report. 2018). However, as discussed in this manuscript, healthy HSCs were shown to do not express EKLF/KLF1 (Frontelo, P. et al. Blood. 2007). This issue was justified based on different cell sources and markers used to identify LT-HSC. However, authors did not use the most updated and standard set of markers to identify primitive HSCs, which includes SLAM markers (see Oguro, H. et al. Cell Stem Cell. 2013). Therefore, it is difficult to assess if sorted cells are pure LT-HSCs or they are mixed with MPP, which are known to express EKLF/KLF1. Authors might consider to refer to CD34-Flk2- HSC instead of LT-HSC in order to bypass this controversy.

- Agree.
We have added descriptions of LT-HSC as defined by different sets of surface markers including the CD34 and Flk2 used in this study of ours.

More importantly, we now clearly indicate that the LT-HSC we have analyzed is Flk2-CD34-HSC, as defined in the Introduction (p.2) and used throughout the manuscript.

2) Considering the previously mentioned role of EKLF/KLF1 in non-physiological HSC, the authors should consider to highlight this aspect in the discussion and cite the corresponding studies.

-Agree.
We have added a paragraph in Discussion (p.9) highlighting these previous findings of the upregulation of EKLF in HSC under non-physiological conditions.

3) In the introduction, sometimes it is not clear if authors are describing general concepts of adult bone marrow hematopoiesis or fetal liver hematopoiesis. Since the latter is the subject of this study, authors might provide some references of studies targeting other molecular/cellular mechanisms specifically in fetal liver hematopoiesis. Similarly, they should clearly state that their claim related to the role of "EKLF as a hierarchical regulator of hematopoiesis in mammals" applies so far to fetal hematopoiesis.

- Agree.
We have rewritten the first two paragraphs of Introduction with clear description and literature citations of hematopoiesis with the emphasis on the process in fetal liver.

4) There are some evidences suggesting that not all EKLF effects are cell autonomous (Porcu, S. et al. Mol Cell Biol. 2011). This might explain why lethally-irradiated recipient mice transplanted with EKLF/KLF1 KO cells manage to survive till around 12 weeks post-transplantation. Since authors use a full KO model in their experiments, they should recognise that HSC phenotype might be partially due to non-cell autonomous mechanisms.

- We have since considered this point on p.8 by highlighting the potential role of non-cell autonomous effect of EKLF in causing the phenotypes we observed in the study.

5) In general figure legends are extremely dense and show lot of information that properly belongs to the Material and Method section, or even to the results section. Please leave the essential information to understand the figures. Also check spaces between words after dots.

- Indeed. We have taken care of this point by shortening a significant portion of the legends, in particular those of Fig. 1 and Fig. 2, some of which are now moved to the Material and Method section.

6) Authors should indicate why mouse erythroleukemia (MEL) cells are used as control for EKLF/KLF1 expression analysis.

- Yes, we should have.
The use of MEL cells as a control is now clarified on p.4.

7) Images provided in Fig. 2C are very small and it is actually difficult to distinguish any difference between WT and KO in terms of CSF2RB expression. Possibly authors could save space omitting merge images since DAPI masks the signal from other channels. In this regard, what do the authors mean when they state that "EKLF staining signals were present in the nucleus" (L.215).

- Agree.
We have now removed the merged images from Figure 2D, enhancing the other images, and also added arrows to indicate where EKLF/ KLF1 molecules are clustered as nuclear bodies in the nucleus of HSC.

8) Authors claim that EKLF/KLF1 KO exhibits enhanced long-term engraftment potential, but hematopoietic reconstitution just 2 months after transplantation is not sufficient for this claim. A minimum of 4 months after transplantation are required for it. Since these mice die after 12 weeks, authors should consider to focus on the self-renewal potential capability assessed by serial transplantations and ignore the effect on the long-term engraftment potential.

- Agree.
We have since removed the phrase "long-term engraftment " on p.6 and p.10.

9) Since the amount of CD45.1- CD45.2- cells significantly changes along serial transplantations (Fig. 3B). Authors should consider to normalize their percentages to the total number of CD45.1 + CD45.2 cells they detect (taking them as 100%). Do they have any explanation to the dramatic decrease in the CD45.2 chimerism of WT cells in tertiary recipients? This is important considering that it increases slightly from primary to secondary recipients.

- Agree.
We have since normalized the % to the total number of CD45.1+CD45.2 cells and presented the normalized % in the new Fig. 2B.

Also, the decrease of CD45.2 chimerism in WT cells in the tertiary recipients might be due to stem cell exhaustion (Roa et al., 2018). This possibility is now highlighted on p.8.

10) Authors fairly recognise that the self-renewal phenotype might be an indirect effect caused by the anemia. However, in the following paragraph they describe it as a consequence of the reduced differentiation capacity of KO HSC, and finally they claim that the role of EKLF on HSC self-renewal still needs to be elucidated (L. 349-354). Please be consistent with the claims.

-The major conclusion from our study is that EKLF/ KLF1 negatively regulates the differentiation of HSC/ HSPC of the hematopoietic system. The reviewer might have mis-understood our wordings in the last paragraph of Discussion.

Please read our writing again. If there is still a question, please let us know.

Minor

1) Please use the same font size and style for the current address information presented in the affiliations section.

- Sorry. This point has been taken care of now;

2) The right nomenclature is Flk2, not F1k2.

- This typo error has now been corrected;

3) L54-56: Provide reference for this claim.

- The reference Pediatrics et al. (2016) is now included in the text;

4) There is no mention in the text to the PCR and IB gels shown in Fig. 1B.

- Sorry about the citation of Fig. 1B before. It is now included on p.3;

5) L96-97: Please clarify what the authors mean with "the hematopoiesis in which was definitive in nature (Lee et al, 2016)".

- We have since removed the sentence "... definitive in nature...." from p.3;

6) Please avoid showing unnecessary mathematical calculations in the main text (e.g page 8). They just add confusion when single numbers are enough.

  • Agree. The mathematical calculations have been removed from p.8.

Please see the attachment for the manuscript.

Reviewer 2 Report

Dear Authors,

The authors report the importance of EKLF/KLF1 role in the differentiation of long-term hematopoietic stem cells. The knockout of EKLF is involved regulating negatively and indirectly this differentiation, thought the Csf2rb/CSF2RB signaling pathway. The English is well written, the design of the experiments is well done.

Minor corrections:

  • Few written typos should be corrected over the manuscript such as Effects (line 336), line 396 (font)
  • Line 350: I will write “strongly suggest” rather than it is clear, because it is not fully clear but strongly suggested that the pathway is involved.

Author Response

  1. Few written typos should be corrected over the manuscript such as Effects (line 336), line 396 (font)

-Sorry about it. These typos have now been corrected;

  1. Line 350: I will write “strongly suggest” rather than it is clear, because it is not fully clear but strongly suggested that the pathway is involved.

-Agree. The wordings have been changed as the reviewer suggested.

Round 2

Reviewer 1 Report

The authors have now addressed my concerns. Please note that the current text requires acceptance of authors edits and delete old figures (current version includes both old and new ones).